# Cryo-EM structure of the human histamine H₁ receptor/G_q complex

Ruixue Xia[1,3], Na Wang[1,3], Zhenmei Xu[1,3], Yang Lu[1], Jing Song[1], Anqi Zhang[2], Changyou Guo[2] & Yuanzheng He [1✉]

Histamine receptors play important roles in various pathophysiological conditions and are effective targets for anti-allergy treatment, however the mechanism of receptor activation remain elusive. Here, we present the cryo-electron microscopy (cryo-EM) structure of the human H₁R in complex with a G_q protein in an active conformation via a NanoBiT tethering strategy. The structure reveals that histamine activates receptor via interacting with the key residues of both transmembrane domain 3 (TM3) and TM6 to squash the binding pocket on the extracellular side and to open the cavity on the intracellular side for G_q engagement in a model of "squash to activate and expand to deactivate". The structure also reveals features for G_q coupling, including the interaction between intracellular loop 2 (ICL2) and the αN-β junction of G_q/11 protein. The detailed analysis of our structure will provide a framework for understanding G-protein coupling selectivity and clues for designing novel antihistamines.

[1] Laboratory of Receptor Structure and Signaling, The HIT Center for Life Sciences, Harbin Institute of Technology, Harbin, China. [2] The HIT cryo-EM facility, Harbin Institute of Technology, Harbin, China. [3] These authors contributed equally: Ruixue Xia, Na Wang, Zhenmei Xu. ✉email: ajian.he@hit.edu.cn

Histamine is a biogenic amine that mediates a variety of pathophysiological responses and signaling events through the binding of histamine receptors, members of the class A G-protein-coupled receptor (GPCR) superfamily[1]. There are four types of histamine receptors, $H_1R$, $H_2R$, $H_3R$, and $H_4R$[2]. $H_1R$ and $H_2R$ are validated targets for the treatment of allergies and forms of gastric acid-related conditions, while $H_3R$ and $H_4R$ have a clinical potential for dementia, asthma, inflammatory bowel disease, and rheumatoid arthritis[1,3–5]. Histamine binding of receptor recruits heterotrimeric G-protein and triggers downstream signaling cascade. The $H_1R$ is mainly coupled with $G_q$ protein that activates phospholipase C to increases inositol phosphates and intracellular calcium level, $H_2R$ couples to $G_s$ protein to stimulate cAMP production, $H_3R$ and $H_4R$ signals through $G_{i/o}$ proteins[2].

Histamine has an active role in allergy and anaphylaxis and is mainly mediated by $H_1R$[3]. The earliest use of antihistamines for allergic disorders can be traced back to early 1950s and over more than half century antihistamines remain as the first choice for many allergic disorders, such as allergic rhinitis, hay fever, and urticarial[3]. Early antihistamines can easily penetrate the blood-brain barrier and have low receptor selectivity, therefore have considerable side effects such as sedation, dry mouth, and arrhythmia[6,7]. The later introduction of carboxyl moiety and protonated amine improve receptor selectivity and significantly reduce side effects associated with brain permeability. However, even the second- or third-generation antihistamine, such as Certirizine (Zyrtec), Loratidine (Claritin), and Fexofenadine (Allegra) still have some unwanted side effects, such as drowsiness, dizziness, and headache[1,8,9]. The most successful antihistamine design is bulky molecules with a basic amino group which is very different from the simple imidazole ring and ethylamine side chain of histamine[3], yet the mechanism by which those bulky antihistamines block the $H_1R$ signaling is still unknown.

An early study has revealed the structure of $H_1R$ bound to the first generation of antihistamine, doxepin, in an inactive conformation[10]. The structure revealed that the amine moiety of doxepin forms a salt bridge with a strictly conserved $D107^{3.32}$ and the bulky tricyclic dibenzooxepin ring sets in a hydrophobic pocket form by conserved residues of TM 3, 5 and 6. The study also explained the improvement of specificity of the second generation of antihistamine via the docking method. However, lacking the active conformation of the receptor, the paper did not articulate a clear mechanism by which those inverse agonists inactivate receptor. Limited structural information of the receptor, particularly the lack of precise agonist binding information and the active conformation of receptor, hampered the development of novel antihistamine that may shut down the receptor activity more effectively and have less side effects.

GPCRs primarily couple to 4 major Gα families, $G_s$, $G_{i/o}$, $G_{q/11}$, and $G_{12/13}$, that dictate different signaling cascade[11]. Considering the large number of receptors (more than 800) and the limited number of G protein (16 of 4 families)[12], a general question is whether there is a selective barcode for receptor/G-protein recognition. A number of studies have suggested that the engagement of G-protein to receptor is more complicated than previous envisioned[13–15], and there is no simple primary or second structural pattern on receptor/G-protein recognition. A barcode, if exists, it must lie in the tertiary or even the quaternary structure of receptor/G-proteins complex. A number of GPCR/$G_s$ and GPCR/$G_i$ complexes have been solved by cryo-EM, however, there is very few of $G_{q/11}$-coupled receptor complex structure available. The muscarinic acetylcholine receptor 1 (M1R)/$G_{11}$ complex structure first revealed some distinct features for the $G_{q/11}$-coupled receptors, including an extended TM5 and a receptor

c-tail/G-protein interaction[16]. Most recently, a 5-$HT_{2A}$ serotonin receptor (HTR2A)/mini-$G_q$ protein complex structure was solved[17]. It is imaginable that with more $G_q$-coupled receptor complex structures being solved, a pattern of receptor/$G_q$ engagement can be found.

Here, we present a cryo-EM structure of $H_1R$ in complex with a N-terminal engineered $G_q$ protein by a NanoBiT tethering strategy. The structure reveals the mechanism of ligand-induced receptor activation in a model of "squash to activate and expand to deactivate" (Supplementary Fig. 1). Our structure analysis also unravels some distinct features of $H_1R$/$G_q$ engagement. These findings could help to understand the $G_q$ coupling selectivity and provide clues for designing novel antihistamines that can more specifically block histamine signaling and have fewer side effects.

## Results

**The overall structure of $H_1R$/$G_q$ complex.** To facilitate the cryo-EM structure solving, we use an engineered $G_q$ protein ($G_{qiN}$) in which the N-terminus (residue 1–32) of $G_q$ was replaced by the N-terminus (residue 1–28) of $G_i$ protein to render the protein binding ability to the scFv16 antibody that has been successfully used in solving numerous receptor/G-protein complexes[16,18–20], including the M1R/$G_{11}$ complex. Also the long ICL3 of $H_1R$ (residue 224–401) was deleted to improve protein expression and folding. The initial attempt to get a stable complex via co-expression of the receptor, $G_{qiN}$ and scFV16 was not very successful. We then adopted to the NanoBiT tethering strategy[21] in which the C-terminus of $H_1R$ was fused to the large part of NanoBiT (LgBiT), and the C-terminus of Gβ was fused to the renovated 13-amino acid peptide of NanoBiT (HiBiT) (Supplementary Fig. 2). This strategy has been successfully used in solving several GPCR/G-protein complexes structures, including the vasoactive intestinal polypeptide receptor[21] (VIP1R)/$G_s$ and growth hormone-releasing hormone receptor/$G_s$ complexes[22]. Indeed, the NanoBiT tethering strategy greatly improves the composition of the complex (Supplementary Fig. 3), and the structure was solved by the single-particle cryo-EM analysis of the $H_1R$-LgBiT/Gα$_{qiN}$/Gβ-HiBiT/Gγ/scFv16 complex at 3.64 Å resolution (Methods and Supplementary Figs. 4–6). The overall reconstruction of the $H_1R$/$G_q$ complex is shown in Fig. 1a with the resulting model in Fig. 1b. The global arrangement of the complex is similar to other GPCR/G-protein complexes with the Gα protein engaging with the intracellular side of the receptor. Local resolution analysis shows that the WD40 repeat domain of Gβ and the core transmembrane domain of the receptor have the highest resolution, the extracellular part of receptor, the border of Gα RAS domain and the N-terminus of Gβγ have the relative lower resolution (Supplementary Fig. 5a). The alpha helical domain of Gα cannot be resolved in our analysis due to its high flexibility. We also observed trace amount of density of the NanoBiT complex formed by the LgBiT and HiBiT (Supplementary Fig. 7), however, the local resolution much worse than 6 Å, we therefore omit it in the later structural analysis. Previous VIP1R study has shown that the fusion of LgBiT to receptor has almost no effect on the function of the receptor. We also did a nuclear factor of activated T-cells response element (NFAT-RE) reporter assay, a well-established reporter assay for the $G_{q/11}$ signaling pathway[23], showing that the fusion of LgBiT to $H_1R$ only marginally affects receptor activity (Supplementary Fig. 2c). Because of the NanoBiT density is far from the receptor/Gq interface (Supplementary Fig. 7) and the fusion of LgBiT do not affect receptor's response to histamine, we reason that the complex structure of $H_1R$-LgBiT/Gα$_{qiN}$/Gβ-HiBiT/Gγ represents the complex structure of $H_1R$/$G_q$, and hence use $H_1R$/$G_q$ to represent the complex in this study. The transmembrane helical bundle has

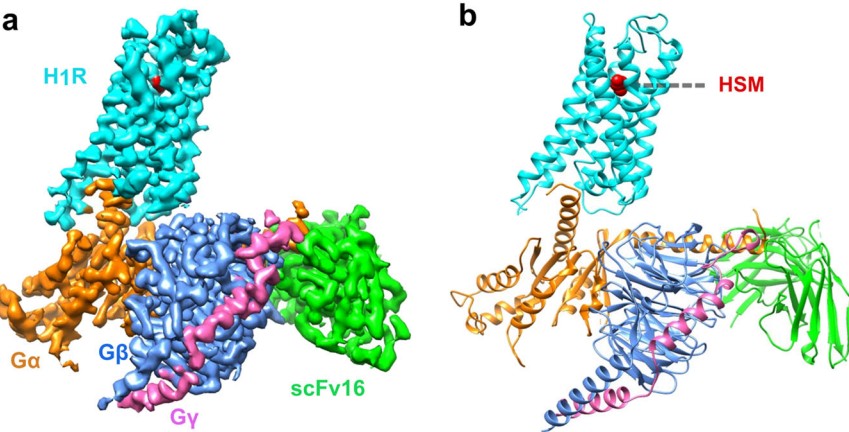

**Fig. 1 Overall structure of the H₁R/G_q complex. a** Orthogonal views of the cryo-EM density map of the H₁R/G_q complex. **b** Model of the complex in same view and color scheme as shown in (**a**). HSM: histamine.

strong electron signal and the histamine ligand is well resolved in the ligand-binding pocket (discussed in detail in the following section).

**Histamine binding and the ligand-binding pocket**. The native ligand histamine is well resolved in the ligand-binding pocket (Fig. 2a). The structure shows that all three histamine nitrogen atoms establish hydrogen bonds with the surrounding polar residues from TM3, TM5, and TM6. The primary amino group of histamine ($N^\alpha$) forms hydrogen bonds with the $D107^{3.32}$ of TM3 and $Y431^{6.51}$ of TM6, the 1 position nitrogen atom ($N^\pi$) of the imidazole ring form a hydrogen bond with $Y431^{6.51}$ of TM6, and the 3 position nitrogen atom ($N^\tau$) forms hydrogen bonds with $N198^{5.46}$ of TM5 and $T112^{3.37}$ of TM3 (Fig. 2a). Those key positions are conserved in the histamine receptor family (Supplementary Fig. 8) and the interactions are confirmed by mutation experiment on the NFAT-RE reporter assay. Single mutation of D107A, N198A, and Y431F totally abolish histamine-induced receptor activation (Fig. 2b), suggesting that they play a key role in ligand binding. The T112A mutation only partially decreases the ligand-induced receptor activity, this may be due to the factor that the $N^\tau$ atom also forms hydrogen bond with the $N198^{5.46}$ of TM5. On the other hand, the Y108F and S111A mutations do not have a detrimental effect on ligand-induced receptor activation, indicating that they do not participate the forming of the hydrogen bond network with the ligand. Interestingly those two residues are also not conserved in the histamine receptor family (Supplementary Fig. 8). The periphery of the ligand-binding pocket is surrounded by hydrophobic residues, including $W428^{6.48}$, $F432^{6.52}$, $F435^{6.55}$, and $W158^{4.56}$ (Fig. 2a). To investigate whether those non-polar bulky residues also contribute to receptor activation through hydrophobic or van der Waals interactions, we mutate those bulky residues (including two polar bulky residues that do not form hydrogen bond with histamine, $Y108^{3.33}$ and $Y458^{7.43}$) to small non-polar residues and test them in the NFAT-RE reporter assay. The data show that mutation of W158A and W428A totally abolish receptor activation, and F435A severely affects receptor activity (Supplementary Fig. 9), suggesting that they may play a crucial role in defining the ligand-binding pocket and providing some necessary hydrophobic interaction to support the correct ligand binding. Particularly, the $W428^{6.48}$ of the conserved CWxP motif is the "toggle-switch" of receptor, plays a key role in switching receptor from inactive state to active state. The Y108V and Y458A mutations retain about half of receptor activity, and the F432A and I454A mutations are almost as active as the wild-type receptor, suggesting those

residues do not directly participate ligand binding. An electro-static potential calculated and analyzed by the APBS Eletrostatics PyMol Plugin shows a negative charged pocket setting on the up-middle (toward the extracellular side) of the receptor (Fig. 2c). The pocket is formed by $D107^{3.32}$, $Y458^{7.43}$, $Y431^{6.51}$, $N198^{5.46}$, $T112^{3.37}$, $Y108^{3.33}$, $S111^{3.36}$, and histamine sets right middle of the pocket.

**Active H₁R vs inactive H₁R**. Compared to the inactive conformation of doxepin-bound H₁R[10] (Cα root-mean-square deviation of 1.235 Å), the biggest difference is the squash of ligand-binding pocket in the active conformation. Calculating by the CASTp 3.0 server[24], the solvent-accessible volume of doxepin-bound H₁R is 249 Å³, and the histamine-bound H₁R is 79 Å³ (Supplementary Fig. 10). A close look at the ligand-binding sites shows that in the active conformation, the agonist histamine forms hydrogen bonds with $D107^{3.32}$, $T112^{3.37}$ of TM3 on one side, and forms hydrogen bond with the $Y431^{6.51}$ of TM6 on the other side, acting as a magnet to pull the extracellular half of TM6 toward TM3 (Fig. 3a). Although the inverse agonist doxepin also forms hydrogen bond with the conserved $D107^{3.32}$ of TM3, its hydrophobic tricyclic dibenzooxepin ring cannot form a hydrogen bond with $Y431^{6.51}$ of TM6, in fact the bulky tricyclic dibenzooxepin ring push $F432^{6.52}$ and $F435^{6.55}$ of TM6 away from TM3 (Fig. 3b). It is conceivable that the contraction of the extracellular binding pocket can lead to the expansion of the intracellular end, thus to open the hydrophobic cavity of the intracellular side for transducer engagement, as seen in β2 adrenergic receptor (β2-AR)/G_s, M1R/G₁₁ and cannabinoid receptor 1 (CB1)/G_i complex[16,25,26]. Indeed, we see an inward displacement of the extracellular part of TM6 (2.6 Å measured by the Cα of $A439^{6.59}$) and a dramatic outward displacement of the intracellular part of TM6 (11.6 Å as measured by the Cα of $N408^{6.28}$) upon ligand binding (Fig. 3a). The conserved $W428^{6.48}$ serves as a pivot point for these movements, where the "up" (extracellular) side move inward (4.6 degree), and the "down" (intracellular) side move outward (20.5 degree) (Fig. 3c). We also see the coordinated movements of TM3, TM2, and TM7 in squashing the binding pocket and opening the hydrophobic cavity of the intracellular side (Fig. 3a). Together, those movements render receptor in an active conformation, enabling the engagement of the αH5 of G_αq. Based on those observations, we postulate that the key mechanism for receptor activation is to squash the ligand-binding pocket via the hydrogen bonds formed by histamine with key residues of TM3 and TM6. To test the hypothesis, we have mutated the $Y431^{6.51}$, the key residue of TM6 that forms hydrogen bond with

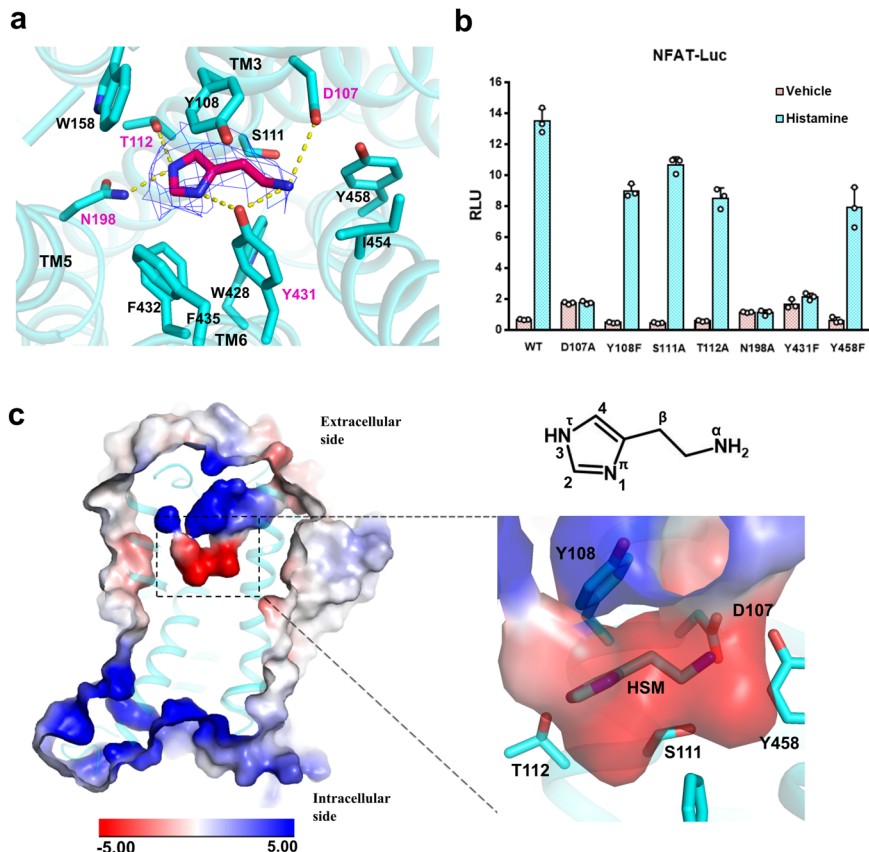

**Fig. 2 Histamine binding and the ligand-binding pocket. a** The ligand-binding pocket of histamine. The histamine is shown in red in the middle of the pocket, density map of histamine (blue mesh) is set at contour level of 3.0. Surrounding residues within 4.1 Å of histamine are shown in sticks and colored in cyan. Hydrogen bonds are marked as yellow dash line. **b** A NFAT-RE reporter assay of mutations of key residues in the ligand-binding pocket. Histamine, 10 μM; data are presented as mean values ± SD; $n = 3$ independent samples. RLU, relative luciferase unit. WT, wild-type. **c** The electrostatic potential surface of $H_1R$ and the histamine binding pocket. The chemical structure of histamine is shown on the up-right panel.

the $N^\pi$ of histamine, to the positive charged residue R, rendering it ability to form a salt bridge with the negative charged residue $D107^{3.32}$ to mimic the effect of histamine to pull TM6 toward TM3 (Fig. 3d). In the NFAT-RE reporter assay, the Y431R shows high basal activity (Fig. 3e). To prove the high basal activity is caused by the salt bridge interaction, we mutated $D107^{3.32}$ to N. The data show that the combination of Y431R/D107N or D107N alone has almost no basal activity. To further prove the lost basal activity of the Y431R/D107N is due to the loss of salt-bridge interaction, we also mutate $D107^{3.32}$ to E which retains the ability to form a salt bridge with Y431R. The data shows while D107E has no basal activity as the D107N, the combination of Y431R/D107E retains the most basal activity of Y131R (Fig. 3e), suggesting the high basal activity indeed comes from the salt-bridge interaction between TM3 and TM6. We also obtained similar results when $Y431^{6.51}$ is mutated to K (Supplementary Fig. 11). Taken together, we propose a model of "squash to activate and expand to deactivate" for $H_1R$ action (Supplementary Fig. 1) where agonist activates receptor via forming hydrogen bonds with TM3 and TM6 to squash ligand-binding pocket on the extracellular side and open the cavity for G-protein engagement on the intracellular side; on the other hand, antihistamine (inverse agonist) use its bulky group to expand the ligand-binding pocket on the extracellular side and to close the G-protein binding cavity on the intracellular side to shut down receptor signaling. Indeed, a comparison of doxepin-bound $H_1R$ with histamine-bound $H_1R$ shows that the bulky tricyclic dibenzooxepin ring of doxepin pushes $F432^{6.52}$ and $F435^{6.55}$ of TM6 away from TM3. It is also

very interesting to find that the most successful antihistamines are those with bulky ring group, such as fexofenadine (Allergra), loratadine (Claritin), and certirizine (Zyrtec) (Supplementary Fig. 12), further supporting the idea of "expand to deactivate and squash to activate". To examine whether this is a specific activation mechanism for $H_1R$, or it is common mechanism for monoamine GPCRs, we compared the size of agonist and antagonist binding pockets of $H_1R$ with β2-AR, dopamine receptor DRD2 and serotonin receptor HTR2A, members of the monoamine GPCRs (Supplementary Fig. 13). The side-by-side comparisons show that the ligand-binding pocket of $H_1R$ condenses most dramatically upon agonist binding, the β2-AR ligand-binding pocket slightly shrinks while there is no significant change in the size of DRD2 pocket upon agonist binding, and HTR2A's pocket actually increase upon agonist binding. Those data suggest that the "squash to activate and expand to deactivate" might be a more specific model for $H_1R$ action.

We also look at how the activation signal on the extracellular side is transmitted to the intracellular side via examining the conformation changes on the conserved DRY, CWxP, NPxxY, and PIF motifs. Those motifs lie on the intracellular half of receptor and are usually associated with locking receptor in the inactive state (Supplementary Fig. 14). The CWxP motif of TM3 lies in the hinge region of receptor where the extracellular side signal is transmitted to the intracellular side of receptor. In receptor activation, the binding of agonist ligand histamine triggers the rotameric switch of $W428^{6.48}$ and the concomitant side chain rotation of $Y431^{6.51}$, $F432^{6.52}$, $F424^{6.44}$ and $F199^{5.47}$, initiating the rotation of TM6.

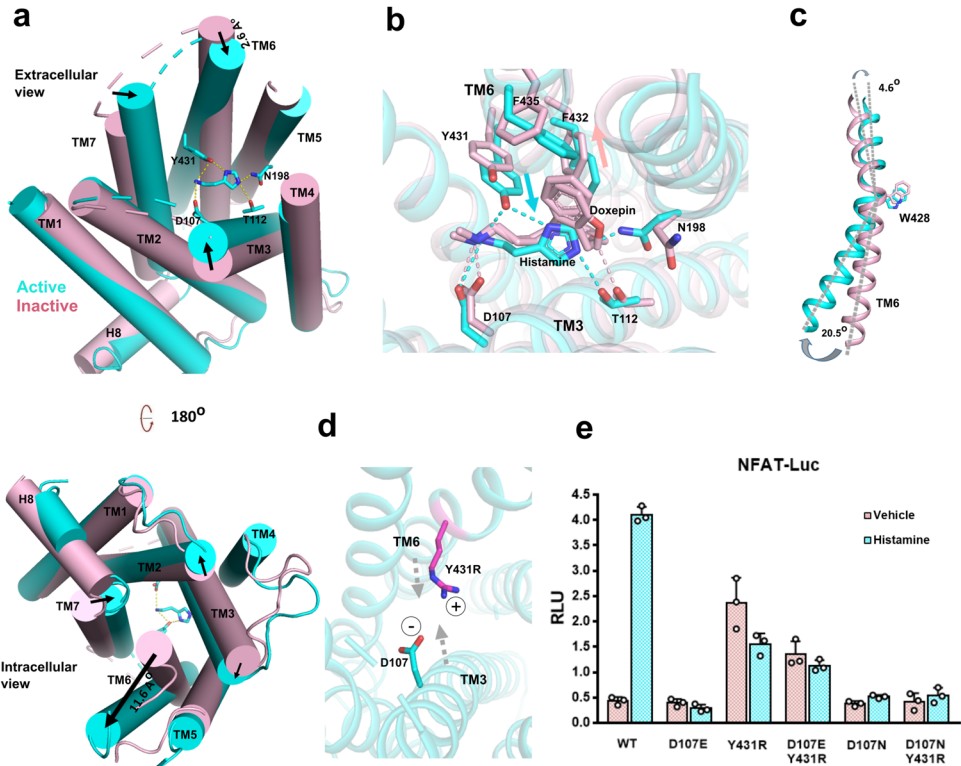

**Fig. 3 Activation of H$_1$R by the binding of histamine. a** A comparison of histamine-bound active H$_1$R and the inverse agonist doxepin-bound inactive H$_1$R (PDB 3RZE). The helices are shown in cylinders. Active receptor is shown in cyan and inactive receptor is shown in light pink. Up-panel is the extracellular side view and lower-panel is the intracellular side view. Arrows mark the movement of the designate part of receptor upon histamine binding. **b** A comparison of the agonist (histamine) binding pocket (cyan) with the inverse agonist (doxepin) binding pocket (light pink). The cyan arrow marks the movements of key residues on TM6 upon agonist (histamine) binding and the light pink arrow marks the movements of key residues on TM6 upon inverse agonist (doxepin) binding. **c** The movements of TM6 upon receptor activation. The W428$^{6.48}$ severs as a pivot between the upside (extracelluar) movement and the downside movement of TM6. **d** A simple model (for illustration only) of a designed salt bridge interaction between the Y431R mutation of TM6 and with the negative charge D107$^{3.32}$ of TM3. **e** A NFAT-RE reporter assay of the Y431$^{6.51}$ and D107$^{3.32}$ mutations. Histamine, 10 μM; data are presented as mean values ± SD; $n$ = 3 independent samples. RLU, relative luciferase unit.

Mutation of W428$^{6.48}$ to A totally abolishes receptor activity (Supplementary Fig. 9b), indicating the importance of this residue in the switch of receptor activity. The DRY motif localizes on the lower part of TM3. In the inactive state, the Y125$^{3.50}$ of DRY motif interacts with both A413$^{6.33}$ and Q416$^{6.36}$ of TM6 to lock receptor in an inactive conformation. Upon receptor activation, the interaction is broken as TM6 sways away from inside to outside (Supplementary Fig. 14d). The N464$^{7.49}$ of the NPxxY motif on TM7 associates with D73$^{2.50}$ in both active and inactive states, however, upon receptor activation, there is 8.2 Å displacement of Y468$^{7.53}$ toward the center of the cavity when TM6 sways away, the aromatic ring of Y468$^{7.53}$ also tilts about 40 degrees to form new contact with residue V118$^{3.43}$, L121$^{3.46}$, and R125$^{3.50}$ of TM3 (Supplementary Fig. 14e). In the PIF motif, the most significant change is the side chain displacement of F424$^{6.44}$ upon receptor activation, similar to the observation in the receptor activation of the HTR2A/mini-Gq complex[17].

**The engagement of G$_q$ to H$_1$R.** The engagement of G$_q$ to H$_1$R is mainly maintained by key interaction from TM6, TM5, TM3, ICL2, and TM7-H8 kink region. The K412$^{6.32}$ of TM6 makes a key interaction with the carboxy group of N357$^{G.H5.24}$ of the αH5 (Fig. 4a). The side chain of N352$^{G.H5.19}$ in αH5 forms a hydrogen bond with the backbone carbonyl of S128$^{3.53}$ in TM3, while the last residue of αH5, Y356$^{G.H5.23}$, is blocked by the R125$^{3.50}$ to prevent further intrusion (Fig. 4b). The ICL2/αN-β1 junction interaction has been reported in M1R/G$_{11}$ complex[16], we also

observe a similar interaction in the H$_1$R/G$_q$ complex but with a different pattern where the backbone carbonyl of Y135$^{ICL2}$ forms a hydrogen bond with the αN-β1 junction. In TM7-H8 kink region, we observe a close contact of N474$^{8.49}$ with the N357$^{G.H5.24}$ of αH5 (Fig. 4a). Residue L133 in ICL2 is highly conserved in GPCR family and has been implicated play an important role in G$_s$ and G$_q$ coupling. Similar to the HTR2A/mini-G$_q$ complex[17], we found that the hydrophobic residue L133$^{ICL2}$ intrudes into a hydrophobic "pocket" formed by L40, V199, F201, F341 and I348 of Gα$_q$ (Supplementary Fig. 15). Mutation studies has suggested that this hydrophobic interaction play a crucial role in stabling the receptor/G$_q$ complex[17]. Interestingly, we also observe a patch of interactions between the ICL1 and the Gβ subunit where the side chain of R56$^{ICL1}$ forms a hydrogen bond with the backbone carbonyl of A309 of Gβ, and the H59$^{ICL1}$ is in close contact with the negative charged D312 of Gβ (Fig. 4c).

The overall engagement of H$_1$R/G$_q$ is similar to M1R/G$_{11}$ (Fig. 4d), however, distinct features were observed. Aligned with receptors, a comparison of M1R/G$_{11}$ with H$_1$R/Gq shows a clear difference in the orientation and position of G protein relative to receptor. The insertion of αH5 into H$_1$R is less deeper (about 1/2 helix) than the G$_{11}$ in M1R (Fig. 4e). Also the αH5 of G$_q$ shows a 9.2 degree outward displacement in the H1R/G$_q$ complex, we also observe the coordinated displacements on Gα RAS domain and Gβγ core domain (Fig. 4d). However, the largest observed displacement is the translational displacement of αN (9.5 Å as measured by the Cα of A7 between G$_q$ and G$_{11}$) when aligned

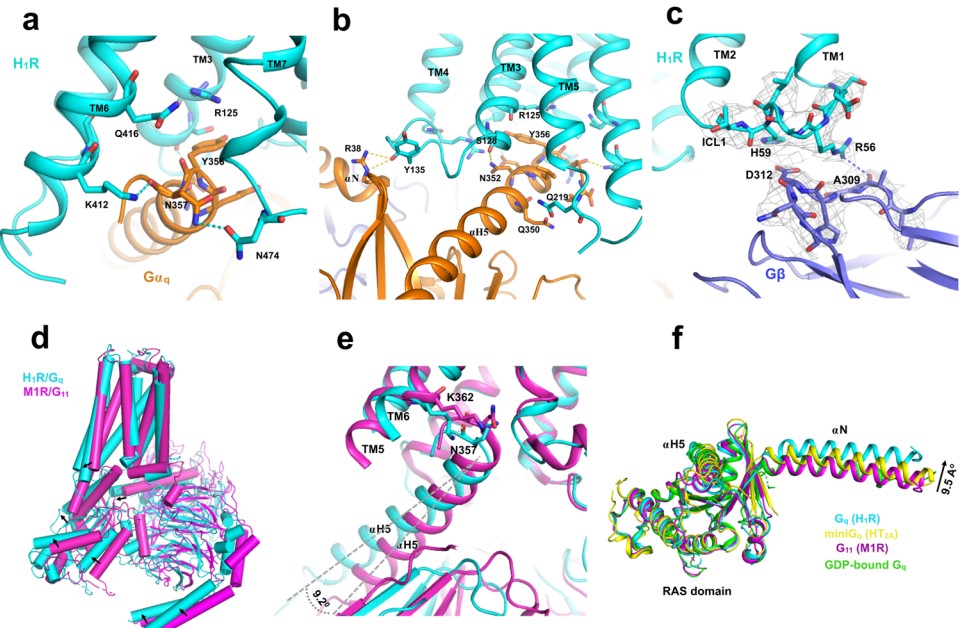

**Fig. 4 The engagement $G_q$ to $H_1R$. a** The interaction between the receptor and the αH5, viewing from the TM6, TM7-H8 front angle. The receptor is colored in cyan and $Gα_q$ is colored in orange. **b** The interaction between the receptor and $Gα_q$, viewing from the TM5, TM3, and ICL2 front angle. **c** the ICL2/Gβ interaction. The receptor is colored in cyan and Gβ is colored in slate. Density map of designated region is set to contour level of 3.0. **d** An overall comparison of the $H_1R/G_q$ complex with the $M1R/G_{11}$ complex (PDB 6OIJ). **e** A comparison of the engagement of αH5 to receptor between the $H_1R/G_q$ complex with the $M1R/G_{11}$ complex. **f** A comparison of the αN displacements of $G_{q/11}$ in receptor engagement. $M1R/G_{11}$ (PDB 6OIJ), HT2AR/mini-$G_q$ (PDB 6WHA) and GDP-bound $G_q$ (PDB 3AH8).

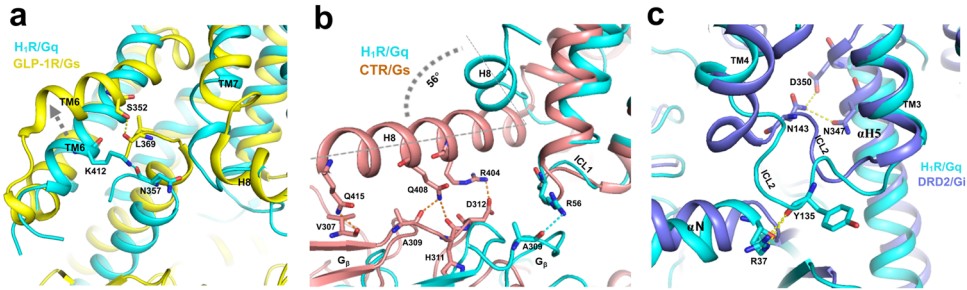

**Fig. 5 Comparisons of $G_q$-coupled $H_1R$ with $G_s$- and $G_i$-coupled receptors. a** A comparison of the αH5 engagement between GLP-1R/$G_s$ (PDB 5VAI) and the $H_1R/G_q$ complex. The dashed arrow shows the additional outward displacement of the TM6 of GLP-1R. **b** A comparison of the Gβ interactions between the CTR/$G_s$ (PDB 5UZ7). **c** A comparison of the ICL2 interaction between the DRD2/$G_i$ (PDB 6VMS) and the $H_1R/G_q$ complex.

with the core of RAS domain which seldom changes shape in receptor engagement (Fig. 4f). The αN of the $H_1R/G_q$ complex shows a dramatic outward translational displacement compared to that of the $M1R/G_{11}$ and HTR2A/mini-$G_q$, as well as the GDP-bound inactive $G_q$ protein[27]. While the interaction between ICL2 and αN-β1 junction seems to be a common feature of the $G_{q/11}$ coupled receptor, $H_1R$ use a different pattern to interact the junction in which the backbone carbonyl of Y135[ICL2] interacts with the side chain of R38 of Gq (Fig. 4b and Supplementary Fig. 16a), instead of the "head to toe" pattern observed in M1R/$G_{11}$ and HTR2A/mini-$G_q$ where the side chain of R134[ICL2] interact with the backbone carbonyl of R37, and the side chain of R37 interact with the backbone carbonyl of R134[ICL2] in the M1R/G11 complex (for HTR2A/mini-$G_q$, the pattern is R185[ICL2]/R32) (Supplementary Fig. 16). We speculate that the interaction difference may contribute to the large displacement of αN in the $H_1R/Gq$ complex. In addition, we also find the TM7-H8 kink/ αH5 interaction might be a common feature for $G_{q/11}$ coupling receptors. In $H_1R/G_q$ complex, the N474[8.49] interacts with the

N357[G.H5.24] of αH5 (Fig. 4a and Supplementary Fig. 17). In HTR2A/mini-$G_q$ complex, the N384[8.47] interacts with N244 of the mini-$G_q$. Similarly in $M1R/G_{11}$, the N422[8.47] of M1R is in a close contact with E355[G.H5.22] and N357[G.H5.24] of the αH5.

**Comparison with GPCR-$G_s$ and -$G_i$ complexes.** The structure of $H_1R/Gq$, $M1R/G_{11}$, and HTR2A/mini-$G_q$ complex, enable us to do a comparison of $G_q$-coupled with $G_s$-coupled and $G_i$-coupled receptor complex. Since Gs complex structures predominantly come from Class-B GPCR/$G_s$ complex, we first compare with the class B GPCR. It has been reported that the $G_s$ family has the most pronounced outward displacement of TM6 in receptor activation. Compared to Glucagon-like peptide 1 receptor (GLP-1R)/$G_s$ complex[28], the TM6 of GLP-1R shows a 6.0 Å more outward displacement than $H_1R$ (Fig. 5a). Similar displacement of TM6 was observed on parathyroid hormone receptor-1 (PTH1R)[29] and calcitonin receptor (CTR)[30]. We also noticed a different pattern of TM6/αH5 interaction. Instead of using the positive charged residue (K412[6.32] of $H_1R$ and

$K362^{6.32}$ of M1R) for αH5 interaction, the Gs-coupled receptor use serine or threonine ($S352^{6.41}$ of GLP-1R and $S409^{6.41}$ of PTH1R) for αH5 interaction (Fig. 5a). It has been reported that the Gβ subunit participates receptor interaction in class B GPCR/G$_s$ complex[22,28–30], since we discovered an interaction between ICL1 and Gβ, we compared H$_1$R with class B GPCRs. We noticed a significant difference in the length and orientation of helix 8, the class B GPCRs have a much longer H8 than the G$_q$-coupling receptor (H$_1$R, M1R and HTR2A). Most notably, a H8 of G$_s$-coupled class B GPCR (CTR) shows a 56 degrees of tilt toward the Gβ subunit side, which explains the general observation of H8/Gβ interaction in this family (Fig. 5b). For instance, the $Q415^{8.67}$, $Q408^{8.6}$ and $R404^{8.56}$ of calcitonin receptor (CTR) make contact with V307, A309, and D312 of Gβ, respectively. On the other hand, H$_1$R use its ICL1 ($R56^{ICL1}$) to interact the Gβ subunit (A309) (Figs. 4c, 5b). The role of H8/Gβ interaction of the class B GPCR is generally believed to be related to receptor stability but not the receptor/G-protein coupling as mutations show little effect on the signaling[28,30]. Consistent with this, mutation of $R56^{ICL1}$ and $H59^{ICL1}$ of the H$_1$R to alanine show no effect on the NFAT-RE reporter assay (Supplementary Fig. 18).

We then compare the class A G$_s$-coupled receptors with H$_1$R. There are only two class A GPCR/G$_s$ complex structures (β2-AR and β1-AR) and two class A GPCR/mini-Gs (GPR52 and A$_{2A}$R) complex structures available. We first compare with Gs complexes. Similar to class B GPCR, the outward displacement of TM6 is more pronounced in β2-AR and β1-AR than H$_1$R (Supplementary Fig. 19a). The TM6/αH5 interaction pattern is similar to class B GPCR, β2-AR and β1-AR use threonine ($T274^{6.36}$ and $T291^{6.36}$, respectively) for αH5 interaction while H$_1$R use positive charged residue ($K412^{6.32}$) for αH5 interaction (Supplementary Fig. 19a). We also noticed a subtle difference of the TM3/αH5 interaction. For instance, β2-AR use both the last two residues ($I135^{3.54}$, $T136^{3.55}$) of TM3 to interact with the αH5 of G$_s$ ($Q384^{G.H5.5.16}$ and $R380^{G.H5.5.12}$) while the G$_q$-coupled receptor only use the second last residue of TM3 ($S128^{3.53}$ of H$_1$R and $S126^{3.53}$ of M1R) to interact the αH5 ($N352^{G.H5.5.19}$ and $Y356^{G.H5.5.23}$) (Supplementary Fig. 19b). For the mini-Gs/Receptor complexes, a distinct feature is the TM5/α H5 interaction ($H233^{5.69}$ of GPR52 interact with Q384 of mini-G$_s$, and $Q207^{5.63}$ of A$_{2A}$R interact with Q374 of mini-G$_s$) which has not been in H$_1$R or M1R (Supplementary Fig. 20).

Compared to the G$_i$-coupled receptors, the difference is much subtle, the most noticeable difference is the ICL2 interaction. The G$_q$-coupled receptors seem uniquely use the ICL2 to interact with the αN-β1 junction of Gα (Fig. 5c and Supplementary Fig. 16a), while the Gi-coupled receptors dominantly use the ICL2-TM4 junction to interact with the αH5. For instance, the D2 dopamine receptor (DRD2) use the $N143^{ICL2}$ to interact with the $D350^{G.H5.22}$ of the αH5 of G$_i$[19] (Fig. 5c), while H$_1$R use the $Y135^{ICL2}$ to interact with the R37 of the αN of Gq. Similarly, CB1[26] use $P221^{ICL2}$, CB2[31] use $K142^{ICL2}$ and μ-opioid receptor (μOR)[20] use $R179^{ICL2}$ to interact with the αH5 of G$_i$ (Supplementary Fig. 21), and M1R use $R134^{ICL2}$ to interact the R37 of the αN of G$_{11}$ (Supplementary Fig. 16a). On the G protein side, compared with G$_s$ and G$_i$-coupled receptor complexes, the αN of H$_1$R/G$_q$ complex displays the largest outward translational displacement (Supplementary Fig, 22) as seen in the G$_q$-coupled receptor complex (Fig. 4f). Mutations of key residues of H$_1$R (Y135A, K137A) that interacts with the αN-β1 junction of G$_q$ shows no severe effect on receptor activity on the NFAT-RE reporter assay (Supplementary Fig. 16b), suggesting that function of the large translational displacement of αN in H$_1$R/G$_q$ complex may not directly relate to receptor coupling.

## Discussion

In this study, we use the NanoBiT strategy to solved the structure of histamine-bound H$_1$R in complex with G$_q$ via single-particle analysis of cryo-EM. To facilitate receptor expression and folding, we use a ICL3 truncation version of receptor (224-401) for structure study. It has been reported that the ICL3 contributes to the recognition and coupling of GPCRs with G$_s$, G$_i$, G$_q$ and G$_{12/13}$[13,14], we therefore ask whether the ICL3 deletion will affect receptor activation. In a NFAT-RE reporter assay, the ICL3 deleted receptor responds ligand well, however, comparing the dose-response curve, the ICL3 truncation shows 10 folders higher of EC$_{50}$ than wild-type receptor (118.6 nM for wild type, 1651 nM for ICL3 truncation, Supplementary Fig. 23), suggesting that the ICL3 does contribute receptor activation and Gq coupling. We suspected that the flexible region of ICL3 may interact with αH5 and the Ras domain of the Gαq that faces the receptor side, and those interactions may stabilize the receptor/Gq complex, the detail of those interaction is worthy of future investigation for a comprehensive understanding of G$_q$ coupling. In summary, we have revealed the active conformation of H$_1$R in complex with G$_q$ and presented a model of "squash to activate and expand to deactivate" for H$_1$R action. Our findings, including both the mechanistic insights and the featured observations of the G$_q$ coupling will benefit both the understanding of G$_q$ signaling and the rational design of novel antihistamines.

## Methods

**Constructs.** The human H$_1$R gene was subcloned into the pFastBac plasmid with a HA-signal peptide sequence on its N-terminus and the LgBiT fused to its c-terminus followed by a Tobacco etch virus (TEV) cutting site and 2 fused maltose-binding proteins to facilitate protein expression and purification. A 12 amino sequence of GASGASGASGAS sequence was inserted between the receptor and LgBiT. The first 28 residues (residue 1-28) and the ICL3 loop (residue 224–401) of H$_1$R were chopped off to increase protein expression and folding. The HiBiT was fused to the c-terminus of human Gβ$_1$ and cloned into pFastBac plasmid as described in the VIP1R paper[21]. The N-terminus (residue 1-32) of human Gα$_q$ was replaced by the N-terminus of G$_i$ (residue 1–28) and subcloned into pFastBac plasmid. The wild-type human Gγ$_2$ was cloned into pFastBac plasmid. The scFv16 that encodes the single-chain variable fragment of mAb16 was subcloned into pFastBac plasmid.

**Expression and purification of H$_1$R/G$_q$ complex.** Bacmid preparation and virus production were performed according to the Bac-to-Bac baculovirus system manual (Gibco, Invitrogen). For expression, the *Spodoptera frugiperda* (Sf9) cells at density of $2 \times 10^6$ cells per ml were co-infected with baculovirus encoding the H$_1$R-LgBiT-tev-2MBP, G$_{qiN}$, Gβ, Gγ and scFv16 protein at a ratio of 1:100 (virus volume vs cells volume). Cells were harvest 48 h after infection. Cell pellets were resuspended in 20 mM Hepes buffer (pH 7.5), 150 mM NaCl, 10 mM MgCl$_2$, 20 mM KCl, 5 mM CaCl$_2$, and homogenized by douncing ~30 times. Apyase was added to the lysis at a final concentration of 0.5 mU/ml. To keep the complex stable, histamine was added at a final concentration of 100 μM all through the purification procedure. The lysate was incubated at room temperature for 1 h with flipping. Then, n-dodecyl-β-D-maltoside (DDM) was added at the final concentration of 0.5% to solubilize the membrane at 4 °C for 2 h. Then the lysis was ultracentrifuged at 56,000g (45,000 rpm) at 4 °C for 40 min. The supernatant was collected and incubated with amylose column for 2 h. The amylose column was washed with a buffer of 25 mM Hepes (pH 7.5), 200 mM NaCl and 0.02% DDM, and 0.004% cholesteryl hemi-succinate (CHS), then eluted with the same buffer plus 10 mM maltose. The elution was concentrated and processed with home-made TEV for overnight at 4 °C. Then the digest was separated on a Superdex 200 Increase 10/300 GL (GE health science) gel infiltration column with a buffer of 25 mM Hepes (pH 7.5), 200 mM NaCl, and 0.1% digitonin (Biosynth). The peak corresponding to the H$_1$R/G$_q$ complex was concentrated at about 10 mg/ml and snap frozen for later cryo-EM grid preparation.

**Grid preparation and cryo-EM data collection.** Three microliters of H$_1$R/G$_q$ complex sample at ~10 mg/ml was applied to a glow-charged quantifoil R1.2/1.3 CuRh holey carbon grids (Quantifoil GmbH). The grids were vitrified in liquid ethane using Vitrobot Mark IV (Thermo Fisher Scientific) instrument in the setting of blot force of 10, blot time of 5 s, humidity of 100%, temperature of 6 °C. Grids were first screened on a FEI 200 kV Arctica transmission electron microscope (TEM) and grids with evenly distributed thin ice and promising grids were transferred to a FEI 300 kV Titan Krios TEM equipped with a Gatan Quantum

energy filter and a spherical corrector for data collection. Images were taken by a Gatan K3 direct electron detector at magnitude of 64,000, super-resolution counting model at pixel size of 0.54 Å, the energy filter slit was set to 20 eV. Each image was dose-fractionated in 32 frames using a total exposure time of 2.56 s at a dose rate of 1.56 e/Å$^2$/s (total dose 50 e/Å$^2$). All image stacks were collected by the EPU program of FEI, nominal defocus value varied from 1.2 to 2.0 µm.

**Data processing.** Raw movies at a size of 0.54 Å were binned once to generate a pixel size of 1.08 Å and then motion-corrected using MotionCor2[32], followed by CTF estimation using CTFFIND 4.1[33]. Particles were picked from the micrographs using crYOLO[34]. Then the picked particles (about 2.4 million) were extracted by RELION[35,36] (version 3.1) and subjected to 2 rounds of reference-free 2D classi-fication in RELION. About 650,000 particles were selected and the initial model was generated by cryoSPARC[37] ab initio. Then the model was used as reference in RELION 3D classification. Classes showed clear secondary structure features were select for a 3D refinement in RELION, followed by a Baysian polishing[38] imple-mented in RELION. Then the polished particles were subject to one round of 2D classification to get rid of the irregular particles that may not contribute to the high resolution 3D reconstitution. Then, this was followed by a 3D refinement and a CTF refinement implemented in RELOIN. The CTF refined particles were sub-jected to a 3D classification with fine angular sampling which yields a promising class of 169,241 particles. Then the particles were transferred to cryoSPARC and followed a Non-uniform Refinement which yields a map of 3.64 Å based on the gold standard Fourier Shell Correlation (FSC) = 0.143 criterion. Later Phenix real_space-refinement show a resolution of 3.3 Å at 0.143. Local resolution esti-mations were performed using either RELION or cryoSPARC.

**Model building.** The crystal structures of human doxepin-bound H$_1$R[10] (PDB 3RZE) and the G$_{11}$ protein complex from the M1R/G$_{11}$ (PDB 6OIJ)[16] were used as initial models for model rebuilding and refinement against the electron microscopy map. All models were docked into the electron microscopy density map using UCSF Chimera[39]. The resulting model was subjected to iterative manual adjust-ment using Coot[40], followed by a rosetta cryoEM refinement[41] at relax model and Phenix real_space refinement[42]. The model statistics were validated using MolProbity[43]. Structural figures were prepared in UCSF Chimera[39] and PyMOL (https://pymol.org/2/). The statistics for data collection and refinement are inclu-ded in Supplementary Table 1.

**The NFAT-reporter assay.** The nuclear factor of activated T-cells response ele-ment (NFAT-RE) reporter assay was performed according to the luciferase reporter assay for deciphering GPCR pathways paper[23] and Promega instruction. Briefly, AD293 cells were split into 24 well plates at a density of 40,000 per well. After one day of growth on 37 °C at 5% CO$_2$, cells (per well) were transfected with 100 ng of NFAT-RE-Luc, 10 ng of pcDNA3-H1R wild-type or mutations, 10 ng of phRGtkRenilla plasmids by X-tremeGENE HP (Roche) at a ratio 3:1 to DNA amount. 16 h after transfection, cells were induced by histamine at 10 µM or vehicle. Six hours after induction, cells were harvested and lysed by addition of 1× Passive Lysis Buffer (Promega), and luciferase activity was assessed by the Dual-Glo Luciferase system (Promega). Data were plotted as firefly luciferase activity nor-malized to Renilla luciferase activity in Relative Luciferase Units (RLU).

**Structure and sequence comparison.** The calculation of the pocket volume was done by the CASTp 3.0 sever[24]. Sequence alignment by the Clustal Omega[44] sever and the representation of sequence alignment was generated using the ESPript[45] website (http://espript.ibcp.fr). The generic residue numbering of GPCR is based on the GPCRdb[46] (https://gpcrdb.org/).

**Reporting summary.** Further information on research design is available in the Nature Research Reporting Summary linked to this article.

## Data availability
Data supporting the findings of this manuscript are available from the corresponding author upon reasonable request. A reporting summary for this article is available as a Supplementary Information file. Source data are provided with this paper. Structural data have been deposited with the PDB (accession number 7DFL), and maps have been deposited with the Electron Microscopy Data Bank (EMDB) with accession numbers EMD-30665.

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

## Acknowledgements

The authors thank the Cryo-EM facility of Harbin Institute of Technology and the Cryo-EM core of the Shuimu Bioscience (Beijing) for sample screening and data collection. We thank the Startup Funds of HIT Center for Life Science. We thank the National Natural Science Foundation of China (32070048 to Y.H.). We thank Dr. Zhiwei Huang for suggestion and support of this project.

## Author contributions

Y.H. conceived the project and design the experiments. R.X. made the expression constructs, purified the proteins and assembled the complex. N.W. made mutation constructs and performed the functional assays. Z.X. purified the proteins and prepared the grids. Y.L. initial the project and made early constructs. J.S. made some mutation constructs. Y.H., A.Z., and C.G. collected the data. Y.H. solved the structure and wrote the manuscript. All authors contributed to data interpretation and preparation of the manuscript.

## Competing interests

The authors declare no competing interests.
