## [Peer Review File · Nature Communications]

Reviewer #1 (Remarks to the Author):

In the manuscript by Xia et al, the authors determined a 3.64 Å cryo-EM structure of the human H1R in complex with an engineered Gq protein via a NanoBiT tethering strategy. This novel structure reveals the recognition mechanism of H1R by histamine. Authors propose a model of “squash to activate and expand to deactivate” through binding pocket size comparison of the active with inactive H1R structure and mutation analysis of pocket residues. This structure also reveals distinct structural features of H1R-Gq protein interface. Together, this novel structure provides structural clues for understanding Gq protein coupling selectivity and peptide recognition and is of great significance for drug design targeting H1R. The structural mechanism for how histamine recognizes and triggers the conformational change of the receptor, as well as Gq coupling, is well analyzed and described.

Major comments:

The authors propose a model of “squash to activate and expand to deactivate” for H1R action. Is this an H1R-specific activation mechanism, or is it shared across monoamine class A GPCRs, such as serotonin or adrenergic receptors? The manuscript will be improved if the authors could compare the binding pocket size of previously reported monoamine GPCRs with pairwise inactive and active structures and add the corresponding discussion in the main text.

The structural model of H1R ICL2 should be more carefully built. For example, a protruding sidechain density towards α N- β 1 junction from ICL2 is ignored. The reviewer speculates that this density belongs to L133, a conserved residue involved in the hydrophobic interaction network between ICL2 and α N- β 1 junction. Please carefully inspect the map of the ICL2 region to make sure all the residues are correctly placed.

ICL3 loop (residue 224-401) of H1R was chopped off for H1R-Gq complex structure determination. Was this truncated ICL3 replaced by any amino acid linker? Additionally, ICL3 was reported to contribute to the coupling of GPCRs with Gs and Gi protein. Does this ICL3 truncation affect Gq coupling activity or histamine-induced activation of H1R?

In the section “Comparison with Gs and Gi coupling,” authors should focus more on structural comparison of the H1R-Gq complex with other Gs-coupled class A GPCRs, not class B GPCRs.

In lines 200-203, the authors state, “The W428^{6.48} forms hydrophobic interactions with surrounding aromatic ring residues, including Y431^{6.51}, F432^{6.52}, F424^{6.44}, and F199^{5.47}.....”. These hydrophobic interactions are not the main determinants to induce the rotation of W428^{6.48}. This sentence might be rewritten as “Histamine triggers the rotameric switch of W428^{6.48} and the concomitant side chain rotation of

F424^{6.44}, initiating the rotation of TM6 of H1R.”

In line 209-210, the authors state, “the aromatic ring of Y468^{7.53} also tilts about 40 degrees....., preventing further intrusion of the α H5 of G α ”. The conformational change of Y468^{7.53} is not to prevent the intrusion of the α H5 of G α . However, it forms new contacts with residues in TM3 (V118^{3.43}, L121^{3.46}, and R125^{3.50}) and enhances the packing of TM3-TM7.

The description of the conformation rearrangement of the conserved PIF motif should also be included in the “Active H1R vs inactive H1R” section.

Minor comments:

The authors should more carefully check the typos and correct minor mistakes in the manuscript, including Ga (line 66), NaboBiT (line 100), NaBiT (line 117), positons (line 131), and TM8-H8 kink (line 215), etc.

In the “Active H1R vs inactive H1R” section, the quoted figures in the main text should be Fig. 3x, not Fig. 2x.

Reviewer #2 (Remarks to the Author):

Manuscript of Xia *et al.*, describes the cryo-electron microscopy (cryo-EM) structure of the human H1R in complex with a N-terminus engineered G_q protein. This is the first structure of the active histamine H1 receptor and by comparing with the inactive structure, authors proposed a possible activation mechanism of the histamine receptor. This is also the first GPCR structure complexed with the full-length G_q protein (although its N-terminus has been engineered). By comparing with the structures of other GPCR-G protein complexes, authors also successfully highlighted the important interactions between the H1 receptor and G_q protein. The paper contains many new findings and should be of great interest to those working on the histamine H1 receptor and/or the structure and function of GPCRs, in general. I, therefore, strongly recommend publishing this paper in Nature Communications. Before publication, however, several important points listed below should be addressed. In addition, the current manuscript contains numerous typos and errors. I have listed some of them, but the language of the manuscript should be checked carefully before publication.

Major points

1. P7, L128. From the figure, the Ca-N-Oh angle is much narrower than 90 deg and a direct hydrogen bond between histamine and Y431 seems unlikely. The authors should

justify why they think this is a direct hydrogen bond but not an interaction through unresolved solvent molecule(s) at this resolution.

2. P7, L132. To show the importance of hydrogen bond network, they mutated polar residues to non-polar ones. They have not, however, mutated non-polar bulky residues to smaller counterparts thus cannot exclude the importance of hydrophobic/van der Waals interactions in addition to hydrogen bonds. The NFAT-RE reporter assays should be performed for the mutants of non-polar bulky residues to strengthen their claim.

3. P7, L141. They claim that the histamine binding pocket is highly negatively charged (“highly negative charged” in the manuscript. It should be fixed). It, however, contains only one negatively charged side chain and the rest are only non-charged polar ones. Authors should explain how they calculated the surface potential of the cavity with emphasis on how they treated partial charges on oxygen and hydrogen atoms.

4. P9, L175 and others. For receptors, usually “basal activity” is used instead of

“self-activity”.

5. P9, L187 and others. Antihistamines are not simple antagonists but are inverse agonists to lock the conformation of the receptor inactive. “Inverse agonist” should be used through the manuscript.

Minor points

6. P6, L111. “the resolution is far beyond 6Å” should be read as “the local resolution is much worse than 6 Å”

0. P11, L215. “interaction with the N357” -> “interaction with the C-terminal carboxy group of N357”. Is this correct? It is very difficult to see this interaction in Fig. 4a and the panel should be improved.

7. P11, L217. “aH5 head residue” -> “the last residue of aH5”

8. P11, L230 and others. You are comparing two different structures. The difference is not a “movement” but a simple displacement. You should change the wording.

9. P11, L234. “engagement.” -> “engagement (Fig.4f). Then, remove Fig.4f at the end of the next sentence.

10. P12, L250. “Comparison with Gs and Gi coupling” -> “Comparison with GPCR-Gs and -Gi complexes”

11. P13, L264. “Gs-coupled Class B GPCRs” -> “a Gs-coupled Class B GPCR (CTR)”

12. Some typos and other small errors in the main text

P6, L117. NaBiT -> NanoBiT

P7, L125. binding pocket. -> binding pocket (Fig. 2).

P8, L155 - P9, L181. Fig. 2 -> Fig. 3

P10, L193. “(Claritin), cetirizine” -> “(Claritin), and cetirizine”

P10, L197. motif -> motifs

P14, L284. aN5 -> aN

P15, L308. was -> were

P15, L315. "processed by" -> "performed according to"

P15, L319. 20 mM Hepes buffer -> 20 mM Hepes buffer (pH 7.5). 20mM KCl, 5 mM CaCl₂, pH7.5. -> 20mM KCl, and 5 mM CaCl₂. ->

P15, L321 and others. lysis -> lysate

P16, L324. at final -> at the final

P16, L327. "a buffer of 25 mM Hepes, pH 7.5, 200 mM NaCl and 0.02% DDM/0.004% cholesteryl hemi-succinate (CHS)" -> "a buffer containing 25 mM Hepes (pH 7.5), 200 mM NaCl, 0.02% DDM, and 0.004% cholesteryl hemi-succinate (CHS)"

P16, L328. "The elution was concentrated and cut with home-made TEV for overnight at 4 °C. Then the cut was separated on..." -> "The elution was concentrated and processed with home-made TEV for overnight at 4 °C. Then the digest was separated on..."

P16, L330. "a buffer of 25 mM Hepes, pH 7.5, 200mM NaCl and 0.1% digitonin (Biosynth)." -> a buffer containing 25 mM Hepes (pH 7.5), 200mM NaCl, and 0.1% digitonin (Biosynth).

P16, L335. A 3 µl -> Three micro liters of

P16, L336. in a -> using

P16, L337. "at setting of blot force of 10, blot time of 5 seconds, humidity of 100%, temperature of 6 °C." -> "in the setting of blot force of 10, blot time of 5 seconds, humidity of 100%, and temperature of 6 °C."

P16, L338 and others. kv -> kV

P16, L339. a promise of high resolution -> promising grids

P17, L350. the crYOLO -> crYOLO

P17, L351. followed -> subjected to

P17, L353. cryoSPARC³⁷ Ab initio -> cryoSPARC³⁷ *ab initio*

P17, L354. "Classes showed a clear secondary structure features and a promise of high resolution were select for a 3D refinement in RELION" -> "Classes showed clear secondary structure features were selected for a 3D refinement in RELION"

P17, L357. contribute -> contribute to

P17, L358. Then followed -> Then, this was followed

P18, L372. in -> using

P18, L377. suggestion -> instruction

P18, L378. CO2 -> CO₂

P18, L381. 10 mm -> 10 mM, 6 hours -> Six hours

P19, L391. “deposited in the PDB with coordinate accession number 7DFL, and” -> “deposited with the PDB (accession number 7DFL), and”

P19, L392. in -> with

14. There are numerous small errors in the figure legends (too many to list for this referee). The language should be checked carefully.

REVIEWER COMMENTS

Reviewer #1 (Remarks to the Author):

In the manuscript by Xia et al, the authors determined a 3.64 Å cryo-EM structure of the human H1R in complex with an engineered Gq protein via a NanoBiT tethering strategy. This novel structure reveals the recognition mechanism of H1R by histamine. Authors propose a model of “squash to activate and expand to deactivate” through binding pocket size comparison of the active with inactive H1R structure and mutation analysis of pocket residues. This structure also reveals distinct structural features of H1R-Gq protein interface. Together, this novel structure provides structural clues for understanding Gq protein coupling selectivity and peptide recognition and is of great significance for drug design targeting H1R. The structural mechanism for how histamine recognizes and triggers the conformational change of the receptor, as well as Gq coupling, is well analyzed and described.

We thank the reviewer for the positive feedback of our work and deeply appreciate his comments.

Major comments:

The authors propose a model of “squash to activate and expand to deactivate” for H1R action. Is this an H1R-specific activation mechanism, or is it shared across monoamine class A GPCRs, such as serotonin or adrenergic receptors? The manuscript will be improved if the authors could compare the binding pocket size of previously reported monoamine GPCRs with pairwise inactive and active structures and add the corresponding discussion in the main text.

We thank the reviewer for such a good suggestion, we have done a side by side comparison of the ligand binding pocket of the monoamine GPCR β 2-AR, DRD2, HTR2A and H₁R (Extended Data Fig.13). The data shows that H₁R condense mostly upon agonist binding, while β 2-AR slightly shrink the binding pocket upon agonist binding, and no change or even reverse change for DRD2 and HTR2A. These data suggest that the “squash to activate and expand to deactivate” might be a more specific model for H₁R. We have added this in our main text.

The structural model of H1R ICL2 should be more carefully built. For example, a protruding sidechain density towards α N- β 1 junction from ICL2 is ignored. The reviewer speculates that this density belongs to L133, a conserved residue involved in the hydrophobic interaction network between ICL2 and α N- β 1 junction. Please carefully inspect the map of the ICL2 region to make sure all the residues are correctly placed.

Thank you very much for pointing this out. We have carefully checked the density map and agree with the reviewer that there should be a protruding sidechain towards the hydrophobic “pocket” formed by L40, F201, V199, F341 and I348 (Extended Data Fig.15). We have correct this and add discussion of the importance of this conserved interaction in our main text.

ICL3 loop (residue 224-401) of H1R was chopped off for H1R-Gq complex structure determination. Was this truncated ICL3 replaced by any amino acid linker? Additionally, ICL3 was reported to contribute to the coupling of GPCRs with Gs and Gi protein. Does this ICL3 truncation affect Gq coupling activity or histamine-induced activation of H1R?

We did not add any linker to ICL3 deletion. We agree that ICL3 has been reported to play important role in coupling G-protein (including Gs, Gi, G_q and G_{12/13}). We have done additional NFAF-RE reporter assay to compare the dose response of wild type and ICL3 deletion receptor, we found while the ICL3 deletion still responds ligand well, the EC50 is much higher than wild type (Extended Data Fig.23), suggesting

that ICL3 does contribute to receptor/Gq coupling. We have add this in the discussion part of main text.

In the section “Comparison with Gs and Gi coupling,” authors should focus more on structural comparison of the H1R-Gq complex with other Gs-coupled class A GPCRs, not class B GPCRs.

We have added comparison of the H1R-Gq complex with other Gs-coupled class A GPCRs, including β 2-AR/Gs, β 1-AR/Gs, GPR52/mini-Gs and A_{2A}R/mini-Gs, please see our main text and Extended Data Fig.19-20.

In lines 200-203, the authors state, “The W4286.48 forms hydrophobic interactions with surrounding aromatic ring residues, including Y431^{6.51}, F432^{6.52}, F424^{6.44}, and F199^{5.47}…….”. These hydrophobic interactions are not the main determinants to induce the rotation of W4286.48. This sentence might be rewritten as “Histamine triggers the rotameric switch of W428^{6.48} and the concomitant side chain rotation of F424^{6.44}, initiating the rotation of TM6 of H1R.”

Thank you very much correcting this. We have made the change on the main text.

In line 209-210, the authors state, “the aromatic ring of Y468^{7.53} also tilts about 40 degrees……, preventing further intrusion of the α H5 of G α ”. The conformational change of Y468^{7.53} is not to prevent the intrusion of the α H5 of G α . However, it forms new contacts with residues in TM3 (V118^{3.43}, L121^{3.46}, and R125^{3.50}) and enhances the packing of TM3-TM7.

Thank you very much correcting this. We have corrected this in main text and redraw the correlated figure and the contact of V118, L121 and R125. (Extended Data Fig.14e).

The description of the conformation rearrangement of the conserved PIF motif should also be included in the “Active H1R vs inactive H1R” section.

Thank you for the suggestion. We have added analysis of the PIF motif in main text and Extended Data Fig.14b.

Minor comments:

The authors should more carefully check the typos and correct minor mistakes in the manuscript, including Ga (line 66), NaboBiT (line 100), NaBiT (line 117), positons (line 131), and TM8-H8 kink (line 215), etc.

Thank you very much for pointing those mistakes out. We have corrected them.

In the “Active H1R vs inactive H1R” section, the quoted figures in the main text should be Fig. 3x, not Fig. 2x.

Thank you so much for pointing out this mistake! We have corrected it.

Reviewer #2 (Remarks to the Author):

Manuscript of Xia *et al.*, describes the cryo-electron microscopy (cryo-EM) structure of the human H₁R in complex with a N-terminus engineered G_q protein. This is the first structure of the active histamine H₁ receptor and by comparing with the inactive

structure, authors proposed a possible activation mechanism of the histamine receptor. This is also the first GPCR structure complexed with the full-length G_q protein (although its N-terminus has been engineered). By comparing with the structures of other GPCR-G protein complexes, authors also successfully highlighted the important interactions between the H₁ receptor and G_q protein. The paper contains many new findings and should be of great interest to those working on the histamine H₁ receptor and/or the structure and function of GPCRs, in general. I, therefore, strongly recommend publishing this paper in Nature Communications. Before publication, however, several important points listed below should be addressed. In addition, the current manuscript contains numerous typos and errors. I have listed some of them, but the language of the manuscript should be checked carefully before publication.

We deeply appreciate such a positive comment on our work! We have analyzed the data and done additional experiments to address the points raised by the reviewer. We also corrected those typos and errors, and modified the language. Thank you.

Major points

1. P7, L128. From the figure, the C(-N-O) angle is much narrower than 90 deg and a direct hydrogen bond between histamine and Y431 seems unlikely. The authors should justify why they think this is a direct hydrogen bond but not an interaction through unresolved solvent molecule(s) at this resolution.

We have measured the angle between the D107-N α -Y431, the angle looks narrow in the fig is because Y431, N α and D107 looks in the same flat surface plane as the imidazole ring in the figure, but when you toggle the structure around a little bit, you will find that the N α is not in the same surface plane as others. We have measured the angle is around 105-110 degree, and is in the range of hydrogen bond, therefore, we think this is a direct hydrogen bond. Current resolution of our map cannot resolve a water other solvent molecule.

2. P7, L132. To show the importance of hydrogen bond network, they mutated polar residues to non-polar ones. They have not, however, mutated non-polar bulky residues to smaller counterparts thus cannot exclude the importance of hydrophobic/van der Waals interactions in addition to hydrogen bonds. The NFAT-RE reporter assays should be performed for the mutants of non-polar bulky residues to strengthen their claim.

Thank you for the suggestion, we have mutated the non-polar bulky residues in the pocket to small non-polar residues and tested them in the NFAT-RE reporter assay, the data shows that W158A, W428A and F435A have a detrimental effect on receptor activation, suggesting that those residues may help define the binding pocket and provide necessary hydrophobic interaction to support the correct ligand binding (Extended Data Fig.9). We have added this discussion in main text.

3. P7, L141. They claim that the histamine binding pocket is highly negatively charged (“highly negative charged” in the manuscript. It should be fixed). It, however, contains only one negatively charged side chain and the rest are only non-charged polar ones. Authors should explain how they calculated the surface potential of the cavity with emphasis on how they treated partial charges on oxygen and hydrogen atoms.

Thank you for correcting this. We have deleted the word of “highly” and modified the sentence accordingly. The surface potential is calculated by the pymol APBS Eletrostatics Plugin program which prepares the molecule by assigning partial charges and adding hydrogens and other missing atoms first, then calculates the electrostatic map and visualizes it.

4. P9, L175 and others. For receptors, usually “basal activity” is used instead of “self-activity”.

Thank you for your suggestion. We have fix this.

5. P9, L187 and others. Antihistamines are not simple antagonists but are inverse agonists to lock the conformation of the receptor inactive. “Inverse agonist” should be used through the manuscript.

Thank you for your suggestion. We have corrected this.

Minor points

6. P6, L111. “the resolution is far beyond 6Å” should be read as “the local resolution is much worse than 6 Å”

Thank you. Fixed

7. P11, L215. “interaction with the N357” -> “interaction with the C-terminal carboxy group of N357”. Is this correct? It is very difficult to see this interaction in Fig. 4a and the panel should be improved.

Yes, we have correct it and redraw the Fig. 4a. to make it clean and clear.

8. P11, L217. “<H5 head residue” -> “the last residue of <H5”

Corrected. Thank you.

9. P11, L230 and others. You are comparing two different structures. The difference is not a “movement” but a simple displacement. You should change the wording.

Fixed, thanks.

10. P11, L234. “engagement.” -> “engagement (Fig.4f). Then, remove Fig.4f at the end of the next sentence.

Fixed, thanks.

11. P12, L250. “Comparison with Gs and Gi coupling” -> “Comparison with GPCR-Gs and -Gi complexes”

Corrected, thanks.

12. P13, L264. “Gs-coupled Class B GPCRs” -> “a Gs-coupled Class B GPCR (CTR)”

Fixed, thank you.

13. Some typos and other small errors in the main text

P6, L117. NaBiT -> NanoBiT

Fixed, thank you.

P7, L125. binding pocket. -> binding pocket (Fig. 2).

Fixed, thank you.

P8, L155 - P9, L181. Fig. 2 -> Fig. 3

Fixed, thank you, appreciated!

P10, L193. “(Claritin), cetirizine” -> “(Claritin), and cetirizine”

Fixed, thank you.

P10, L197. motif -> motifs

Fixed, thank you.

P14, L284. <N5 -> <N

Fixed, thank you.

P15, L308. was -> were

Fixed, thank you.

P15, L315. “processed by” -> “performed according to”

Fixed, thank you.

P15, L319. 20 mM Hepes buffer -> 20 mM Hepes buffer (pH 7.5). 20mM KCl, 5 mM CaCl₂, pH7.5. -> 20mM KCl, and 5 mM CaCl₂. ->

Fixed, thank you.

P15, L321 and others. lysis -> lysate

Corrected, thanks.

P16, L324. at final -> at the final

Corrected, thanks.

P16, L327. “a buffer of 25 mM Hepes, pH 7.5, 200 mM NaCl and 0.02% DDM/0.004% cholesteryl hemi-succinate (CHS)” -> “a buffer containing 25 mM Hepes (pH 7.5), 200 mM NaCl, 0.02% DDM, and 0.004% cholesteryl hemi-succinate (CHS)”

Corrected, thanks.

P16, L328. “The elution was concentrated and cut with home-made TEV for overnight at 4 °C. Then the cut was separated on...” -> “The elution was concentrated and processed with home-made TEV for overnight at 4 °C. Then the digest was separated on...”

Corrected, thanks.

P16, L330. “a buffer of 25 mM Hepes, pH 7.5, 200mM NaCl and 0.1% digitonin (Biosynth).” -> a buffer containing 25 mM Hepes (pH 7.5), 200mM NaCl, and 0.1% digitonin (Biosynth).

Corrected, thanks.

P16, L335. A 3 µl -> Three micro liters of

Fixed, thank you.

P16, L336. in a -> using

Fixed, thank you.

P16, L337. “at setting of blot force of 10, blot time of 5 seconds, humidity of 100%, temperature of 6 °C.” -> “in the setting of blot force of 10, blot time of 5 seconds, humidity of 100%, and temperature of 6 °C.”

Fixed, thank you.

P16, L338 and others. kv -> kV

Fixed, thank you.

P16, L339. a promise of high resolution -> promising grids

Fixed, thank you.

P17, L350. the crYOLO -> crYOLO

Fixed, thank you.

P17, L351. followed -> subjected to

Fixed, thank you.

P17, L353. cryoSPARC₃₇ Ab initio -> cryoSPARC₃₇ *ab initio*

Fixed, thank you.

P17, L354. “Classes showed a clear secondary structure features and a promise of high resolution were select for a 3D refinement in RELION” -> “Classes showed clear secondary structure features were selected for a 3D refinement in RELION”

Fixed, thank you.

P17, L357. contribute -> contribute to

Fixed, thank you.

P17, L358. Then followed -> Then, this was followed

Fixed, thank you.

P18, L372. in -> using

Fixed, thank you.

P18, L377. suggestion -> instruction

Fixed, thank you.

P18, L378. CO2 -> CO₂

Fixed, thank you.

P18, L381. 10 μm -> 10 μM, 6 hours -> Six hours

Fixed, thank you.

P19, L391. “deposited in the PDB with coordinate accession number 7DFL, and”

-> “deposited with the PDB (accession number 7DFL), and”

Fixed, thank you.

P19, L392. in -> with

Fixed, thank you.

14. There are numerous small errors in the figure legends (too many to list for this referee). The language should be checked carefully.

Thank you, we have checked and modified the language. We deeply appreciate your suggestion to improve the quality and the accuracy of our paper!

Reviewer #1 (Remarks to the Author):

All my concerns have been addressed in the revised manuscript.

Reviewer #2 (Remarks to the Author):

I am very happy about their revision of the manuscript and this should be accepted to be published in Nature communications.